# Can Large Language Models Understand Intermediate Representations in Compilers?

Hailong Jiang [* 1]   Jianfeng Zhu [* 1]   Yao Wan [2]   Bo Fang [3]   Hongyu Zhang [4]   Ruoming Jin [1]   Qiang Guan [1]

## Abstract

Intermediate Representations (IRs) play a critical role in compiler design and program analysis, yet their comprehension by *Large Language Models* (LLMs) remains underexplored. In this paper, we present an explorative empirical study evaluating the capabilities of six state-of-the-art LLMs—GPT-4, GPT-3, DeepSeek, Gemma 2, Llama 3, and Code Llama—in understanding IRs. Specifically, we assess model performance across four core tasks: *control flow graph reconstruction*, *IR decompilation*, *code summarization*, and *execution reasoning*. While LLMs exhibit competence in parsing IR syntax and identifying high-level structures, they consistently struggle with instruction-level reasoning, especially in control flow reasoning, loop handling, and dynamic execution. Common failure modes include misinterpreting branching instructions, omitting critical operations, and relying on heuristic reasoning rather than on precise instruction-level logic. Our findings highlight the need for IR-specific enhancements in LLM design. We recommend fine-tuning on structured IR datasets and integrating control-flow-sensitive architectures to improve the models' effectiveness on IR-related tasks.[1]

## 1. Introduction

Intermediate Representations (IRs) play a pivotal role in compiler design by structuring the compilation process into front-end, middle-end, and back-end phases (Reissmann et al., 2020; Webb et al., 2021; Sbîrlea et al., 2015). Likewise, IRs enable efficient transformations, optimizations, and analyses that are independent of specific programming languages, thereby promoting portability across different architectures (Chow, 2013).

Beyond their role in compilation, IRs are crucial for a variety of code intelligence tasks, including *vulnerability detection* (Zhu et al., 2024; Jiang et al., 2024b), *code comprehension and generation* (Yuan et al., 2023; Jiang et al., 2024c), *clone detection* (Paul et al., 2024), and *binary-to-source matching* (Mao et al., 2023; Collyer et al., 2023). Unlike high-level programming languages, IRs adopt an instruction-level format that encodes low-level details such as control flow dependencies, instruction transformations, and memory operations. These characteristics introduce unique challenges for *structural analysis*, *syntactic processing*, *semantic understanding*, and *execution reasoning*, making IR comprehension fundamentally distinct from modeling natural language or high-level source code.

Current IR processing methods, such as NCC (VenkataKeerthy et al., 2020), represent IR instructions as linear sequences of tokens using Word2Vec-style embeddings (Church, 2017), and BERT-style pretraining (Devlin, 2018; VenkataKeerthy et al., 2020; Niu et al., 2024). However, these approaches lack an understanding of control flow structures and execution semantics, limiting their ability to generalize to IR-specific tasks such as *Control Flow Graph* (CFG) reconstruction, bug detection, and execution reasoning. Since IR execution is governed by low-level operational semantics rather than surface syntax, models that fail to capture control flow relationships often struggle with execution-aware IR tasks.

**Our Study**   Recently, with the remarkable advancements in LLMs for source code understanding and generation (Jiang et al., 2024d; Wang et al., 2023; Liu et al., 2023a), a natural question arises: *Can LLMs understand IRs in compilers?* To investigate this, we systematically evaluate LLMs' comprehension of IRs by assessing their performance across three dimensions. An overview of our evaluation framework and its alignment with the RQs is illustrated in Figure 1.

We first investigate *whether LLMs can accurately recon-*

---

[*]Equal contribution [1]Kent State University, USA [2]Huazhong University of Science and Technology, China [3]Pacific Northwest National Laboratory, USA [4]Chongqing University, China. Correspondence to: Yao Wan <wanyao@hust.edu.cn>, Bo Fang <bo.fang@pnnl.gov>, Qiang Guan <qguan@kent.edu>.

*Proceedings of the 42nd International Conference on Machine Learning*, Vancouver, Canada. PMLR 267, 2025. Copyright 2025 by the author(s).

[1]All the experimental data and source code are publicly available at https://github.com/hjiang13/LLM4IR.

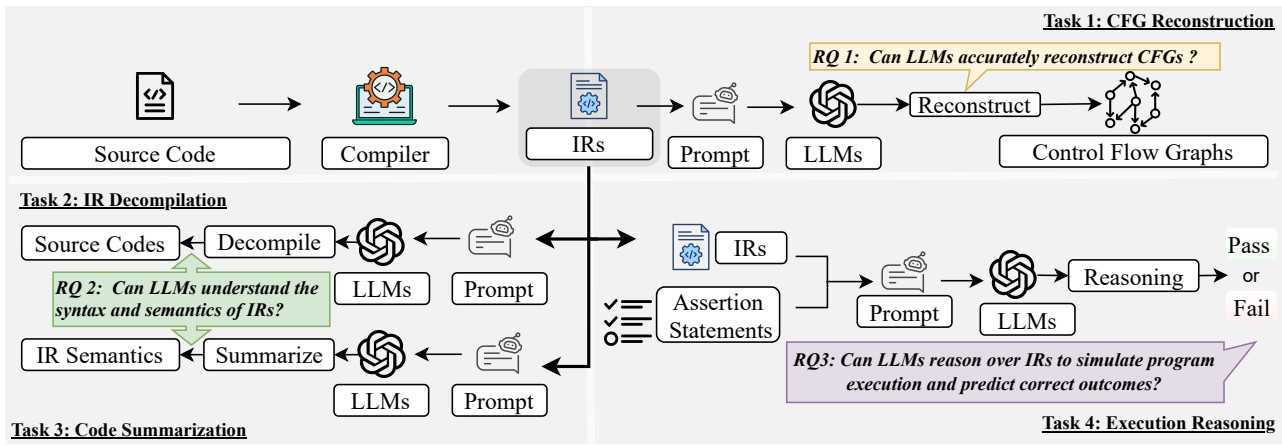

*Figure 1.* Overview of our evaluation framework for assessing LLM understanding of IRs across four tasks.

*struct CFGs from IRs* (RQ1). To evaluate structural understanding (RQ1), we design Task 1 to prompt LLMs to reconstruct CFGs from a LLVM IR. This directly tests their ability to identify basic blocks, interpret control-flow instructions (e.g., branches and loops), and infer execution paths—keys for IR analysis and compiler optimization.

We further examine *whether LLMs understand the syntax and semantics of IRs* (RQ2). This RQ targets both syntactic recognition and semantic comprehension, which are critical for understanding IR beyond its surface form. We evaluate this through two tasks: decompilation (Task 2: converting IR into high-level source-like code) and semantic summarization (Task 3: generating natural language descriptions of IR behavior).

Lastly, we investigate *whether LLMs can reason over IRs to simulate program execution and predict correct outcomes* (RQ3). This RQ probes the models' capacity for dynamic reasoning—tracking control flow, variable states, and instruction effects under specific inputs. We design an execution reasoning task where LLMs are provided with IR code and input values, and are asked to determine whether assertions on output states will hold (Task 4).

In our experiments, we evaluate six state-of-the-art LLMs: GPT-4 (OpenAI, 2023), GPT-3 (Brown et al., 2020), DeepSeek (Guo et al., 2024), Llama 3 (Touvron et al., 2023), Gemma 2 (Team et al., 2024), and Code Llama (Rozière et al., 2024). All evaluations are conducted on a benchmark dataset derived from HumanEval (Zheng et al., 2023), consisting of 164 C++ programs paired with their corresponding LLVM IRs. Each program is compiled using Clang at four optimization levels—-O0, -O1, -O2, and -O3—to generate a diverse set of LLVM IRs that capture both unoptimized and progressively optimized code structures.

**Takeaway Findings** Our results reveal several key insights into how LLMs process IRs:

- RQ1 (Task 1): LLMs can recover basic blocks and partial graph topology, but often misconstruct control flow edges—especially in branches and loops—revealing limited structural reasoning in CFG reconstruction.
- RQ2 (Tasks 2-3): LLMs can parse IR syntax and identify basic structural patterns, but frequently skip critical instructions or rely on heuristic approximations when generating code or summaries—indicating a semantic gap in IR understanding.
- RQ3 (Task 4): LLMs approximate execution reasoning through high-level semantic steps instead of simulating instruction-level behavior, leading to failures in tracking control flow, variable updates, and iteration logic—even with chain-of-thought prompting.

**Contributions** The primary contributions of this paper are summarized as follows.

- To the best of our knowledge, this is the first study on LLMs' capacity to comprehend and reason over IRs, covering structural analysis, syntactic and semantic understanding, and execution reasoning.
- We empirically evaluate state-of-the-art LLMs across multiple IR-processing tasks, introducing targeted tasks, rigorous evaluation metrics, and systematic analyses.
- Our findings pinpoint critical limitations—such as insufficient control flow awareness, shallow instruction-level semantics, and flawed execution reasoning—laying the foundation for future IR-specific model designs.

## 2. Preliminaries

**LLVM and IRs** LLVM is a widely used compiler infrastructure that optimizes code via IRs (Lattner & Adve, 2004). LLVM IRs are *low-level, statically typed*, and follow the

*Static Single Assignment (SSA)* form, offering a language-agnostic representation. They support control flow analysis and enable custom compiler passes.

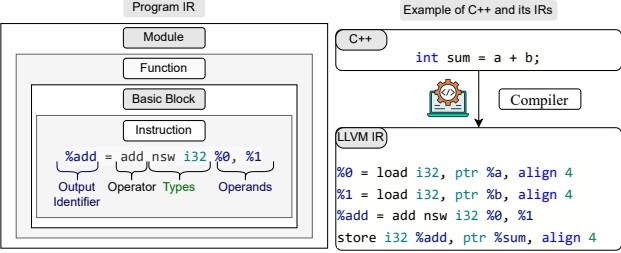

*Figure 2.* An illustration of LLVM IR with the corresponding hierarchical structure.

Figure 2 shows the hierarchical structure of a program's LLVM IR, from modules down to individual instructions. An LLVM IR program consists of several hierarchical components. At the top level, a *module* encapsulates a set of related functions, serving as a compilation unit. Each *function* contains a sequence of *basic blocks*, which are straight-line instruction sequences with a single entry and a single exit. These basic blocks form the nodes in control flow graphs and are the fundamental units of control transfer. Inside each block, *instructions* define operations with explicit opcodes, operand types, and values, governed by LLVM's strict static typing system.

**Control Flow Graphs (CFGs)**   A CFG is a fundamental program representation that captures all possible execution paths of a program. Each node in the CFG corresponds to a basic block—a straight-line sequence of instructions with a single entry and exit point—while edges represent possible control transfers between these blocks. CFGs serve as the backbone for numerous compiler and analysis tasks. In vulnerability detection, they help identify unreachable code, malicious branches, and control hijacks (Zhou et al., 2019; Anju et al., 2010). In compiler optimizations, CFGs enable transformations such as dead code elimination, loop unrolling, and branch prediction (McConnell, 1993). Additionally, in broader program analysis, CFGs support symbolic execution, slicing, and structural reasoning (Fechete et al., 2008), making them indispensable for understanding and manipulating program semantics at the IR level.

## 3. Study Design

As shown in Figure 1, to address our RQs (RQ1–3), we design four evaluation tasks (Task 1–4), each targeting a specific dimension of IR comprehension and aligned with the reasoning skills required to answer the corresponding research question.

### 3.1. Evaluation Tasks

**CFG Reconstruction (Task 1)**   This task evaluates whether LLMs can accurately reconstruct the CFG from a given IR function. As RQ1 focuses on structural understanding, this task serves as a direct probe: generating a CFG necessitates interpreting branches, loops, and control transfers—key components of IR structure. A model that successfully reconstructs the CFG demonstrates a substantive grasp of the structural semantics inherent in the code.

**IR Decompilation (Task 2)**   To investigate whether LLMs understand the syntax of IR (RQ2), we evaluate their ability to translate IR back into high-level source code. Decompilation serves as a rigorous test of syntactic comprehension, as it demands the recognition of structural patterns—such as loops and conditionals—from a flattened IR format, and their accurate reconstruction into readable, high-level constructs. A successful decompilation indicates that the model has a sufficient grasp of IR syntax to recover valid source-level structures.

**Code Summarization (Task 3)**   This task complements Task 2 by shifting the focus toward semantic understanding. In this setting, LLMs are required to generate concise natural language summaries of IR functions. Addressing RQ2 involves more than surface-level parsing; it demands a deeper comprehension of the IR's behavior and purpose. Summarization serves as a proxy to evaluate whether the model can accurately capture functional intent, behavior, and output—ultimately revealing its capacity to reason about function-level semantics.

**Execution Reasoning (Task 4)**   To address RQ3, we evaluate whether LLMs can simulate the execution of IR given specific input values. This task requires the model to determine whether assertions on the output state hold, necessitating accurate tracking of control flow, data dependencies, and variable updates. Unlike simple pattern matching, execution reasoning demands that the model interpret IR as a dynamic process, reflecting a deeper level of semantic understanding.

Together, these tasks provide a comprehensive and structured benchmark aligned with the three core RQs, offering insights into how well LLMs can understand, interpret, and reason over compiler IRs.

### 3.2. Studied LLMs

In this paper, we investigate six state-of-the-art LLMs that span a range of model families, training objectives, and application scopes. These models are selected to provide a balanced coverage of proprietary and open-source models, general-purpose and code-specialized LLMs, as well as diverse training paradigms (e.g., instruction tuning, multimodal pretraining), ensuring a representative and fair evaluation across usage scenarios.

▷ **GPT-4** (OpenAI, 2023) and **GPT-3** (Brown et al., 2020), developed by OpenAI have demonstrated strong performance across a wide range of code-related tasks. GPT-4, in particular, is known for its enhanced reasoning capabilities and broader context window.

▷ **DeepSeek R1** (Guo et al., 2024) is an open-source, instruction-tuned model trained on both code and natural language corpora. Notably, it comes with built-in support for Chain-of-Thought (CoT) prompting, enabling multi-step reasoning without manual CoT demonstrations.

▷ **Llama 3** (Touvron et al., 2023), developed by Meta, is a family of high-performance transformer models with open weights.

▷ **Code Llama** (Rozière et al., 2024) extends Llama with additional pretraining on source code, making it more capable in code synthesis and static analysis.

▷ **Gemma 2** (Team et al., 2024) is an instruction-tuned model released by Google, offering competitive performance in general-purpose and technical reasoning tasks. These models cover a spectrum of training objectives, access modes, and design philosophies, providing a comprehensive basis for evaluating LLMs' ability to reason over compiler IRs.

### 3.3. Datasets

Following (Zheng et al., 2023; Tan et al., 2024), we use the HumanEval benchmark, which comprises 164 programming tasks introduced by OpenAI (Chen et al., 2021). Each task includes a function signature, natural language description, reference implementation, and an average of 7.7 assertion statements. In our experiments, the C++ functions serve as the basis for generating LLVM IRs across various compiler optimization levels. For further discussion on dataset generalizability and task complexity, please refer to Appendix B.

Table 1 summarizes the key IR characteristics across different optimization levels from `-O0` to `-O3`. At `-O0`, the IR remains verbose due to the lack of optimizations, preserving duplicated functions such as `linkonce_odr` to facilitate debugging. As the optimization level increases to `-O1` and `-O2`, redundant structures are pruned, and moderate loop constructs begin to emerge as a result of early optimization passes. The `-O3` level applies more aggressive transformations that substantially restructure the IR, introducing deeper and more complex loop-level optimizations.

### 3.4. Prompt Design

Effective prompt design is crucial for optimizing LLM performance (Liu et al., 2022). Traditional single-turn prompts that embed role instructions and task descriptions into a single input often lack clarity and structured guidance. Recent studies (Liu et al., 2023b; Park et al., 2023) demonstrate that LLMs perform more reliably when treated as *expert agents*

*Table 1.* Key statistics of HumanEval IRs under different optimization levels (`-O0` to `-O3`). BB: Basic Block.

| *opt* Level | LoC | Tokens | Functions | Loops | BBs |
|---|---|---|---|---|---|
| `-O0` | 162,389 | 1,079,290 | 5,195 | 463 | 13,228 |
| `-O1` | **66,598** | **454,954** | 393 | **577** | **7,708** |
| `-O2` | 69,102 | 478,002 | 374 | 625 | 8,048 |
| `-O3` | 75,519 | 526,395 | **367** | 827 | 8,917 |

and are given detailed, domain-specific instructions.

To enhance response precision and consistency, we adopt an **Expert Meta-Template Prompt** format:

---
**Prompt**

*You are an expert in [relevant field].*

**Input Data:** [IR snippet or relevant context]

**Task:** [Task description]

**Output Format:** [Expected output structure]

---

This template ensures that LLMs receive explicit role assignments, domain knowledge, task objectives, and output formatting requirements—collectively guiding them toward more accurate and well-structured responses.

For each of the four tasks (i.e., CFG reconstruction, IR decompilation, code summarization, and execution reasoning), we iteratively refine prompts using strategies such as few-shot learning and CoT prompting (Wei et al., 2022; Xie et al., 2025). Multiple prompt variants are evaluated to maximize accuracy, consistency, and task alignment.

### 3.5. Implementations

**Task 1: CFG Reconstruction** We prompt LLMs to generate CFGs from IR snippetsusing a structured meta-template that defines graph syntax and node semantics (see Section 3.4). The prompt includes explicit guidance and iterative refinement. Following (Ma et al., 2024), we evaluate CFG reconstruction along multiple axes. *Node construction accuracy* measures the correctness of identified basic blocks, while *edge construction accuracy* evaluates the accuracy of control-flow edges. Based on these, we define two aggregate metrics. *Full CFG accuracy* refers to the case where both nodes and edges match the ground truth, whereas *partial CFG accuracy* applies when nodes are correct but edges contain errors. To capture finer-grained structural reasoning, we additionally report *loop edge accuracy*, which measures correct reconstruction of loop structures (e.g., `for`, `while`), and *conditional branch accuracy*, which evaluates the correctness of conditional branches (e.g., `if-else`).

**Task 2: IR Decompilation** We prompt LLMs to decompile IR into C++ source code and compare the results generated

to the original ground truth. The task probes whether LLMs can recover syntactically valid and semantically faithful source code from IRs, across optimization levels (O0–O3). Following prior work (Tan et al., 2024; Armengol-Estapé et al., 2024; Wong et al., 2023; Jiang et al., 2024a), we evaluate decompiled outputs using three hierarchical metrics. *task completions* assesses whether the model produced a syntactically valid code. *execution completion*, checks whether the code compiles and runs. Among these, we distinguish between *re-execution success* (all assertions pass) and *re-execution mismatch* (at least one assertion fails).

**Task 3: Code Summarization** We prompt LLMs with IR code snippets and request function-level summaries. To improve output quality, we apply expert-designed meta-prompts with few-shot examples and, where appropriate, CoT guidance. The generated summaries are compared to ground-truth or human-curated references. Following (Wu et al., 2024), we evaluate summary quality using BLEU (Papineni et al., 2002) for surface fluency, ROUGE-L (Lin, 2004) for content coverage, and METEOR (Denkowski & Lavie, 2014) for semantic alignment.

**Task 4: Execution Reasoning** We present LLMs with IR snippets along with embedded assertion statements and prompt them to predict which assertions evaluate to "*true*." This task requires the model to simulate program behavior through static reasoning over control flow and variable assignments. We evaluate execution reasoning by checking assertion-level correctness: a *pass* if all are correct, *partial pass* if some are correct, and *failure* if none are. We also report the *overall pass rate*, the proportion of correctly predicted assertions across all cases.

## 4. Experimental Results and Analysis

To assess how effectively LLMs understand IRs, we structure our analysis around the three RQs outlined in Figure 1. This section presents the quantitative results and highlights key findings. Each model output is categorized as either *Task Completed*, indicating that the model produced a valid response, or *Task Failed*, typically due to hangs or crashes caused by IRs exceeding the model's context window.

### 4.1. Structural Understanding (RQ1)

In this RQ, we evaluate whether LLMs can reconstruct CFGs from IRs, revealing their structural understanding. We assess LLMs' structural understanding by prompting them to reconstruct CFGs from IRs. Their outputs are compared with compiler-generated ground truth using metrics including full and partial CFG accuracy, as well as correctness of loop and conditional edges. Table 2 presents model-wise performance across 164 applications, ensuring robustness through multiple runs.

*Table 2.* LLM performance on CFG construction.

| LLMs | Task Comp. | Node Acc. | Full Acc. | Partial Acc. Loop | Condition |
|---|---|---|---|---|---|
| GPT-4 | **164** | 50 | 39 | **9** | **2** |
| GPT-3 | 133 | 18 | 14 | 3 | 1 |
| DeepSeek R1 | 77 | **64** | **57** | 6 | 1 |
| Gemma 2 | 98 | 18 | 16 | 2 | 0 |
| Llama 3.1 | 76 | 17 | 15 | 2 | 0 |
| Code Llama | 67 | 0 | 0 | 0 | 0 |

### 4.1.1. STRUCTURAL REASONING OVER CFGS

From Table 2, we observe substantial variation in both task completion and CFG generation accuracy across LLMs. GPT-4 completes all 164 cases but produces only 39 fully correct CFGs. In contrast, DeepSeek R1 handles just 77 cases yet yields 57 accurate CFGs, suggesting stronger per-instance reasoning capabilities. This improvement is likely attributable to its use of CoT prompting, which explicitly guides step-by-step control flow reasoning. Other models, including GPT-3 (14 correct), Gemma 2 (16), and Llama 3.1 (15), show weaker performance, while Code Llama fails to generate any correct CFGs.

Structural errors further confirm these limitations. GPT-4 alone incurs 114 node-level mistakes, with GPT-3 and Gemma 2 each exceeding 100. Loop and conditional edge recovery remains poor across the board: GPT-4 reconstructs only 9 loops and 2 branches; DeepSeek R1 achieves 6 and 1, respectively, despite fewer inputs. These results highlight persistent difficulties in accurately identifying instruction boundaries and modeling control dependencies.

> **Finding 1:** LLMs can identify basic control instructions and construct partial graph structures, but often misrepresent branches and loops—revealing limited structural reasoning and poor control dependency understanding. Chain-of-thought prompting improves accuracy by enabling stepwise CFG reconstruction.

### 4.1.2. IMPACT OF GRAPH DENSITY

Upon further analysis, we observe that applications in different performance groups exhibit distinct density patterns, suggesting that density influences model behavior and reconstruction outcomes.

**Graph Density** Graph density quantifies how interconnected a CFG is, calculated as:

$$\text{Density} = \frac{\text{Number of Edges}}{n \times (n-1)} \qquad (1)$$

where $n$ is the number of nodes.

*Table 3.* The number of applications in each category for different LLMs in decompilation tasks, including task completion, re-execution, and success across optimization levels (-O0–-O3).

| LLMs | Task Completed | | | | Re-execution Completed | | | | Re-execution Success | | | |
|---|---|---|---|---|---|---|---|---|---|---|---|---|
| **Opts** | -O0 | -O1 | -O2 | -O3 | -O0 | -O1 | -O2 | -O3 | -O0 | -O1 | -O2 | -O3 |
| GPT-4 | **161** | **164** | **164** | **164** | **34** | 34 | **38** | 32 | **15** | 12 | 10 | 14 |
| GPT-3 | 95 | 145 | 144 | 133 | 2 | **7** | 7 | 4 | 2 | **4** | 4 | 0 |
| DeepSeek R1 | 62 | 76 | 75 | 77 | 31 | **39** | **40** | **38** | 22 | **16** | 17 | 17 |
| Gemma 2 | 72 | 100 | 100 | 98 | 0 | 12 | 16 | 9 | 0 | 7 | 6 | 6 |
| Llama 3.1 | 72 | 97 | 96 | 77 | 0 | 24 | 26 | 23 | 0 | 15 | 15 | **14** |
| Code Llama | 65 | 88 | 83 | 120 | 14 | 16 | 21 | 19 | 4 | 6 | 6 | 4 |

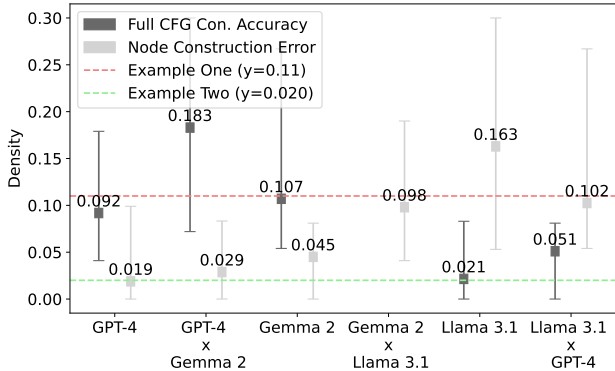

*Figure 3.* Density metrics comparing Complete CFG Accuracy vs. Node Construction Error.

Figure 3 compares complete CFG accuracy (black bars) and node construction error (gray bars) across density levels. The $x$-axis groups models by overlapping task success (e.g., "GPT-4 only" or "GPT-4 × Gemma 2"), while the $y$-axis indicates the average density of those programs. Error bars denote the density range per category. The distinct density patterns between successfully reconstructed CFGs and those with high node errors indicate that graph density strongly correlates with LLM performance. This insight can inform model selection. As shown by the two examples in the Figure 3, CFG density appears to influence which models succeed or fail on the task:

- Higher-density CFGs (avg. 0.11): GPT-4 and Gemma 2 succeed more often, while Llama 3.1 fails.

- Lower-density CFGs (avg. 0.02): Llama 3.1 performs better, while GPT-4 and Gemma 2 struggle.

> **Finding 2:** LLM performance on CFG reconstruction varies with graph density, indicating its potential as a model selection heuristic in IR analysis.

## 4.2. Syntax and Semantic Comprehension (RQ2)

In RQ2, we evaluate whether LLMs can comprehend both the syntactic and semantic aspects of IRs. We address this via IR decompilation and code summarization (Tasks 2-3).

### 4.2.1. HEURISTIC IR RECONSTRUCTION

We use the IR decompilation task to test syntactic understanding, where LLMs are prompted to reconstruct high-level C++ code from IR. Outputs are evaluated based on syntactic validity, execution behavior, and correctness of predicted assertions (Tan et al., 2024; Armengol-Estapé et al., 2024; Wong et al., 2023). All experiments are conducted across optimization levels (-O0 to -O3) to test robustness under compiler-induced IR variation.

Table 3 shows the number of applications in each category for different LLMs in decompilation task. From Table 3, most task failures occur at -O0, where minimally optimized IRs are the longest (Table 1), often exceeding model context limits. GPT-4 handles these cases best with only 3 failures, while Code Llama fails on 99 applications. Interestingly, re-execution success peaks at -O1 and -O2, likely because they preserve structured control flow while reducing redundancy—making IRs easier for LLMs to interpret. In contrast, -O3's aggressive optimizations (e.g., loop unrolling, instruction reordering) obscure control dependencies and degrade decompilation accuracy.

Across multiple examples, we observe that LLMs frequently omit segments of IR—such as entire `if` statements nested within loops—resulting in incomplete or oversimplified decompiled code. Instead of following a strict instruction-by-instruction reconstruction process, LLMs often rely on heuristic pattern recognition to infer missing logic. This heuristic behavior limits their capacity to capture fine-grained control flow semantics. In particular, loop structures are often approximated rather than faithfully reproduced, leading to missing branches or partially handled conditions. Similar patterns are evident in code summarization and execution reasoning tasks (Section 4.3), where LLMs fail to

*Table 4.* Performance comparison of LLMs in the code summarization task, including BLEU (B), ROUGE-L (R-L), METEOR (M).

| LLMs | Task Comp. | High-Quality | | | Avg. Scores | | |
|---|---|---|---|---|---|---|---|
| | | B | R-L | M | B | R-L | M |
| GPT-4 | **164** | 1 | **9** | **21** | 0.35 | 0.56 | 0.63 |
| GPT-3 | 131 | 1 | 1 | 1 | 0.24 | 0.47 | 0.56 |
| DeepSeek R1 | 49 | 1 | 5 | 10 | **0.42** | **0.64** | **0.705** |
| Gemma 2 | 88 | 1 | 4 | 7 | 0.35 | 0.52 | 0.63 |
| Llama 3.1 | 81 | 1 | 5 | 11 | 0.39 | 0.61 | 0.67 |
| Code Llama | 123 | 1 | 1 | 1 | 0.25 | 0.43 | 0.55 |

*Table 5.* Execution reasoning performance of LLMs across IRs and Source Code (SC). Pass rates are calculated based on the number of completed tasks.

| LLMs | Task Comp. | | Pass | | Partial Pass | | Pass Rate | |
|---|---|---|---|---|---|---|---|---|
| | IRs | SC | IRs | SC | IRs | SC | IRs | SC |
| GPT-4 | 164 | 164 | **59** | 118 | 102 | 44 | 0.36 | 0.72 |
| GPT-3 | 164 | 164 | 7 | 26 | **135** | **135** | 0.04 | 0.158 |
| DeepSeek R1 | 164 | 164 | 32 | 119 | 132 | 25 | 0.195 | **0.804** |
| Gemma 2 | 164 | 164 | 32 | 100 | 132 | 63 | 0.195 | 0.610 |
| Llama 3.1 | 164 | 164 | 31 | 119 | 114 | 35 | 0.189 | 0.73 |
| Code Llama | 163 | 164 | 44 | **131** | 119 | 33 | **0.270** | 0.80 |

fully represent execution semantics. As shown in Figure 5 (Appendix F.2), such semantic drift can result in outputs that pass re-execution checks while omitting critical logic.

> **Finding 3:** LLMs can partially generate syntactically valid C++ code from IR, but often rely on heuristic reconstruction, skipping critical instructions and misrepresenting control flow—revealing limited semantic understanding.

### 4.2.2. PATTERN-BASED SUMMARIZATION

To evaluate LLMs' ability to capture the broader semantics of IR, we prompt each model to generate natural language descriptions of IR-derived code snippets (Task 3). This task assesses whether LLMs can articulate a function's purpose, inputs, outputs, and internal behavior. Generated summaries are compared to predefined golden references using BLEU, ROUGE-L, and METEOR. Higher similarity scores indicate stronger semantic comprehension and the ability to infer meaningful program behavior.

Table 4 presents model performance on Task 3. GPT-4 completes all 164 summaries, while GPT-3, Gemma 2, Llama 3.1, and Code Llama fail on 31, 76, 83, and 41 programs, respectively. However, high-quality summaries (score > 0.8) are rare across all models. Each produces only one such case in BLEU. For ROUGE-L, GPT-4 leads with 9 high-scoring outputs, followed by Llama 3.1 (5), Gemma 2 (4), and others (1 each). GPT-4 also ranks highest in METEOR with 21 cases, ahead of Llama 3.1 (11) and Gemma 2 (7). In terms of average scores, DeepSeek R1 performs best overall, achieving the highest BLEU (0.42), ROUGE-L (0.64), and METEOR (0.705). Llama 3.1 and GPT-4 follow closely, while GPT-3 and Code Llama lag behind, reflecting weaker semantic modeling. These results highlight that while LLMs can generate summaries, their ability to capture fine-grained IR semantics remains limited.

These results suggest that while LLMs can identify surface-level syntactic features—such as function signatures, in-

put/output types, and simple arithmetic—they struggle with deeper semantic elements. The scarcity of high-quality summaries (score > 0.8 across BLEU, ROUGE-L, and METEOR) indicates difficulty in capturing iteration patterns, conditional branches, and nontrivial data dependencies. This gap implies that current models rely more on pattern recognition than on reconstructing true program logic. One representative example is provided in Appendix F.3.

> **Finding 4:** LLMs demonstrate an initial understanding of IR syntax and semantics, yet their outputs often omit critical instructions, oversimplify control flow, and rely on shallow patterns—revealing limited sensitivity to fine-grained semantics and an incomplete grasp of underlying program logic.

### 4.3. Execution Reasoning (RQ3)

In this RQ, we aim to evaluate whether LLMs can simulate the behavior of IR code, reasoning over control flow, variable state, and memory updates to predict runtime outcomes with certain inputs.

We prompt LLMs with IR snippets annotated with assertion statements, derived from the HumanEval benchmark, and ask them to predict which assertions evaluate to `true`. To encourage detailed reasoning, prompts request intermediate execution logs, tracing control flow decisions, and variable states. We further design a counterfactual setup, where models are given both IR and the corresponding source code, to compare execution behavior across representations. Outputs are evaluated using pass/fail metrics based on the correctness of assertion predictions.

### 4.3.1. APPROXIMATION IN EXECUTION REASONING

Table 5 summarizes LLM performance on execution reasoning for both IR and source code. All models complete most tasks successfully, with only one failure from Code Llama on an IR case—indicating that execution reasoning likely processes IR incrementally, step by step, rather than all at

once as in structural or syntactic tasks. This interaction style may alleviate token limitations seen in earlier tasks. Despite this, a clear performance gap remains: LLMs consistently perform better on source code, suggesting they are more attuned to high-level constructs. GPT-4 achieves the highest IR pass count (59), while Code Llama leads on source code (131 passes), likely benefiting from code-specific pretraining. These trends imply that IR-specific training may be essential to close the gap between SC and IR reasoning.

In this task, we observe that LLMs often struggle with complex control flows, leading to inconsistencies between IR and source code execution. While most models handle simple control logic correctly, assertion failures increase with nested conditions or iterative patterns. For instance, in "CPP_93", an encoding function that replaces vowels works for a short input (`encode("TEST") == "tgst"`), but fails on a longer one (`encode("I DoNt KnOw WhAt tO WrItE")`), due to skipped transformation steps during reasoning.

These failures reveal that LLMs do not simulate execution instruction by instruction. Instead, they approximate behavior using high-level semantic steps. Analysis of reasoning logs shows that models typically organize their execution into abstract phases—such as function understanding, input analysis, operation planning, and assertion comparison—rather than processing each instruction sequentially. This abstraction introduces gaps when reasoning over control dependencies, particularly in loops or conditionals.

We also observe granularity issues. LLMs can identify overall algorithmic intent (e.g., sorting), but miss critical details. In "CPP_149", where the program deletes strings of odd length before sorting, the model correctly recognizes sorting but omits the filtering step—changing the outcome.

> ***CPP_149 Function Description:***
> *A function that accepts a vector of strings as a parameter deletes the strings that have odd lengths from it, and returns the resulting vector in sorted order.*

However, the LLM skipped an 'odd number removal" step, altering execution behavior. Here is a reasoning log snippet:

> ***Step 3:*** *Analyzing the LLVM IR code:*
>
> *- The LLVM IR code is complex and involves multiple operations, including comparisons, memory allocations, and calls to other functions.*
>
> *- Key parts of the code involve loading strings from the vector, comparing them, and potentially invoking sorting or filtering operations.*

This further demonstrates the limitations of high-level approximations and motivates the need for more structured, step-tracking reasoning mechanisms, such as fine-tuning on execution traces or reinforcement-guided prompting.

> **Finding 5:** LLMs approximate program execution through high-level semantic steps rather than faithfully simulating instruction-level behavior, leading to failures in complex control flows.

### 4.3.2. HEURISTIC INFERENCE UNDER UNCERTAINTY

Unlike source code, IR lacks explicit variable names and high-level semantics, making execution reasoning more ambiguous. When faced with such uncertainty, LLMs rarely express a lack of understanding. Instead, they often fall back on heuristic strategies—such as inferring behavior from function names or assertion patterns—rather than reasoning from instruction semantics.

This behavior frequently leads to incorrect outcomes. In 35 of 164 GPT-4 cases, we observe such pattern-based guessing. For example, in "CPP_120", a function named `maximum` leads the model to assume that the IR computes top-$k$ elements, even without evidence from the actual IR instructions. Similarly, in "CPP_109", the model infers that a vector can be sorted through shifts simply because the assertion passes, without reasoning through the IR-defined shift operations.

These failures highlight LLMs' inability to handle uncertainty in IR reasoning. Even with CoT prompting, models fail to track control flow, variable updates, or simulate intermediate steps. As a result, they often prioritize plausibility over correctness, revealing a fundamental limitation in operational semantic understanding. These behaviors reveal a potential limitation in how LLMs manage uncertainty during IR reasoning—a pattern further discussed in Section 5.

> **Finding 6:** When facing ambiguity in IR execution, LLMs often rely on heuristics, such as function names or output patterns, rather than simulating operations, revealing challenges in semantic alignment.

## 5. Discussion

Our evaluation reveals fundamental challenges in LLMs' ability to understand and reason over compiler IRs, particularly in control flow, execution semantics, and iterative logic. These difficulties stem from both the intrinsic properties of IRs, such as their verbosity and lack of high-level abstractions, and the limitations of current LLM architectures.

IRs typically produce longer token sequences per function compared to source code, intensifying context length constraints. This not only limits the model's ability to retain global context but also disrupts its handling of structural dependencies. Even with full context, models frequently

fail on tasks that require precise execution tracking or fine-grained semantic reconstruction. GPT-4 outperforms other models across tasks, consistent with prior findings on code understanding (Ma et al., 2023), but still falls short on complex control and semantic reasoning.

These findings reveal three key limitations in current LLMs and point to corresponding opportunities for improvement. First, control flow comprehension is fragile—LLMs frequently misrepresent branching and looping structures, leading to degraded structural fidelity. Second, models often rely on superficial pattern matching rather than simulating execution. In the face of ambiguity, they tend to infer logic from function names or assertion syntax instead of expressing uncertainty or reasoning through operations, highlighting a lack of uncertainty modeling in IR reasoning. Third, loop reasoning remains a major weakness: models struggle to track induction variables, identify termination conditions, and capture inter-iteration dependencies. Addressing these limitations may require instruction-level fine-tuning, supervision via execution traces, or prompting strategies that explicitly incorporate uncertainty.

## 6. Related Work

**LLMs for High-level Programming Languages**  Recent advancements in LLMs for NLP have extended to code understanding, enabling models to comprehend high-level languages like Python, C++, and Java (Wan et al., 2024). Models such as GPT-3 (Brown et al., 2020), GPT-4 (OpenAI, 2023), Llama (Touvron et al., 2023), and Claude 3 (Anthropic, 2024) excel at tasks like code generation, translating natural language into executable code. Specialized models, including StarCoder (Li et al., 2023), Code Llama (Rozière et al., 2024), and Code Gemma (Team, 2024), enhance these capabilities, tackling more complex coding tasks and driving advances in software engineering applications (Zhao et al., 2023; Wan et al., 2024).

**IR Representation Learning**  IR representation learning integrates structural and flow-based features, such as token sequences (Peng et al., 2021; Gui et al., 2022), SAT (Guo et al., 2023), CFGs (VenkataKeerthy et al., 2020; Yu et al., 2020; Chu et al., 2024), and control-data flow graphs (CD-FGs) (Ben-Nun et al., 2018; Brauckmann et al., 2020; Cummins et al., 2021). In terms of model architectures, graph neural networks have been widely used to encode the structures of CFGs and CDFGs via message-passing techniques (Brauckmann et al., 2020; Cummins et al., 2021; Yu et al., 2020). Other approaches include skip-gram embeddings, such as inst2vec (Ben-Nun et al., 2018), and relation-based embeddings, like TransE (Bordes et al., 2013), which are trained on CDFGs and CFGs to produce instruction-level embeddings. However, these models lack task-agnostic pre-trained embeddings, limiting their ability to capture essential contextual information for downstream tasks. Approaches like IR2Vec (VenkataKeerthy et al., 2020) address this limitation by introducing hierarchical vector representations that enhance the semantic understanding of IRs. Recent work, such as FAIR (Niu et al., 2024), further refines IR representations using graph Transformers to reduce over-smoothing issues while incorporating pre-training tasks that explicitly capture IR token semantics and flow-type information. Meta's LLM Compiler (Cummins et al., 2024) aligns with these efforts, offering pre-trained models for code optimization tasks. While previous work has explored IR representation for code optimization, no study has systematically examined how LLMs understand IR syntax, CFGs, execution behavior, and semantics. This study provides the first empirical evaluation of LLMs' IR comprehension across these dimensions.

## 7. Conclusion and Future Work

This study systematically evaluates the capabilities of LLMs in understanding IRs in compilers, across four dimensions: structural analysis, syntax comprehension, semantic reasoning, and execution simulation. Our findings show that while LLMs can effectively recognize static IR features and basic control flow patterns, they struggle with more complex constructs—particularly loop reasoning, precise control dependencies, and instruction-level execution behavior. Furthermore, LLMs often omit key instructions during decompilation, misinterpret control structures, and approximate execution in coarse semantic steps rather than performing fine-grained simulation. This contrasts with their strong performance on other code tasks, suggesting a lack of exposure to IR and the need for IR-specific training.

To advance IR-level understanding, future work should enhance control flow awareness, support multi-granularity semantic reasoning, and improve loop modeling. Furthermore, incorporating IR-specific training, optimizing token-efficient representations, and leveraging graph-based neural architectures may further strengthen LLMs' reasoning over IR. Bridging the gap between source code comprehension and IR analysis is crucial for applying LLMs to compiler design, program analysis, and binary code understanding.

## Acknowledgments

This research was supported by the U.S. Department of Energy, through the Office of Advanced Scientific Computing Research's "End-to-end co-design for performance, energy efficiency, and security in AI-enabled computational science (ENCODE)" project. Pacific Northwest National Laboratory is operated by Battelle Memorial Institute for the U.S. Department of Energy under Contract No. DE-AC05-76RL01830.

## Impact Statement

This work investigates LLMs' ability to understand IRs, advancing research in code analysis, optimization, and automated reasoning. It may benefit software reliability, but it also raises concerns about automated program manipulation in security-sensitive contexts. We advocate for further ethical safeguards in applying LLMs to such domains.

Several limitations merit attention. Our study covers a limited set of models and HumanEval-derived IRs, which may not fully reflect the diversity of real-world IRs. While we employ expert meta-prompts, we do not explore advanced prompting techniques, which could affect performance. Moreover, though we identify key challenges in IR comprehension, we do not propose concrete solutions such as IR-specific fine-tuning or dataset augmentation. Future work should expand datasets, include newer models, and explore targeted adaptation strategies to improve LLM performance on both static and dynamic IR tasks.

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

# A. Comprehensive Related Work

**LLMs for High-level Programming Languages** Advancements in pre-trained LLMs for natural language processing (NLP) have significantly extended to code understanding, enabling models to comprehend high-level programming languages such as Python, C++, and Java. General-purpose models like GPT (Brown et al., 2020), GPT-4 (OpenAI, 2023), Llama (Touvron et al., 2023), and Claude 3 (Anthropic, 2024) have shown strong performance in tasks such as code generation, translating natural language descriptions into executable code. Meanwhile, specialized models such as StarCoder (Li et al., 2023), Code Llama (Rozière et al., 2024), DeepSeek-Coder (Guo et al., 2024), and Code Gemma (Team, 2024) have further refined these capabilities, tackling more complex programming tasks and supporting a range of software engineering applications (Zhao et al., 2023).

Pre-trained LLMs have transformed NLP by learning generalizable language representations from large-scale corpora, which can be fine-tuned for diverse downstream tasks (Qiu et al., 2020). Early embedding models such as Word2Vec (Mikolov, 2013) and GloVe (Pennington et al., 2014) captured semantic similarity but lacked contextual sensitivity (Han et al., 2021). The emergence of transformer-based architectures, exemplified by GPT (Radford et al., 2019) and BERT (Devlin et al., 2018), enabled context-aware modeling, substantially improving performance on language understanding tasks (Vaswani et al., 2017).

These LLMs have been adapted to the programming domain, leading to specialized models such as CodeBERT (Feng et al., 2020), GraphCodeBERT (Guo et al., 2020), UnixCoder (Guo et al., 2022), and CodeT5 (Wang et al., 2021). These models integrate Transformer architectures to encode code semantics effectively. Enhancements such as data flow integration in GraphCodeBERT (Guo et al., 2020), multi-modal learning in UnixCoder (Guo et al., 2022), and encoder-decoder frameworks in CodeT5 (Wang et al., 2021) enable improved code comprehension and generation.

**IR Representation Learning** IR representation learning incorporates structural and flow-based features such as token sequences (Peng et al., 2021), CFGs (VenkataKeerthy et al., 2020; Yu et al., 2020), and control-data flow graphs (CDFGs) (Ben-Nun et al., 2018; Brauckmann et al., 2020; Cummins et al., 2021).

For model architectures, graph neural networks (GNNs) have been widely employed to encode CFG and CDFG structures through message-passing techniques (Brauckmann et al., 2020; Cummins et al., 2021; Yu et al., 2020). Other strategies include skip-gram embeddings, such as inst2vec (Ben-Nun et al., 2018), and relation-based embeddings, such as TransE (Bordes et al., 2013), trained on CDFGs and CFGs to generate instruction-level embeddings. However, these models lack task-agnostic pre-trained embeddings, preventing them from capturing contextual information crucial for downstream tasks.

Approaches like IR2Vec (VenkataKeerthy et al., 2020) mitigate this issue by introducing hierarchical vector representations to improve semantic comprehension of IRs. Recent work, such as FAIR (Niu et al., 2024), further refines IR representations using graph Transformers to reduce over-smoothing issues while incorporating pre-training tasks that explicitly capture IR token semantics and flow-type information. FAIR has achieved state-of-the-art performance across multiple code-related tasks, highlighting the increasing importance of pre-trained IR models.

Meta's LLM Compiler (Cummins et al., 2024) aligns with these efforts, offering pre-trained models for code optimization tasks. While prior work has explored IR representation learning for code optimization and analysis, no studies have systematically examined how LLMs comprehend IR syntax, CFG structures, execution behavior, and semantic relationships. Our study addresses this gap by providing the first empirical evaluation of LLMs' IR comprehension across these dimensions.

# B. Discussion on Dataset Scope and Generalizability

Although the HumanEval dataset consists of relatively short source programs (averaging fewer than 10 lines of Python code), their corresponding LLVM IRs exhibit substantial structural and semantic complexity. We selected this dataset to investigate fundamental challenges in LLM-based IR comprehension under controlled conditions. Our rationale is based on the following considerations:

- **Controlled Complexity.** Despite the brevity of HumanEval source programs, their compiled LLVM IRs contain nontrivial structures—including nested loops, conditional branches, function calls, and memory operations—that typify core IR-level reasoning challenges. This controlled environment allows us to isolate and assess LLMs' ability to understand structural and semantic aspects of IRs.
- **Task Difficulty Beyond Code Size.** The difficulty of IR reasoning does not scale linearly with source code length. Even short programs can produce intricate IR patterns that demand nuanced comprehension of control flow, data dependencies,

and low-level semantics. Our results show that LLMs often fail to capture these subtleties, despite the apparent simplicity of the original code.

- **Evidence of Systemic Deficiencies.** Across multiple tasks—including CFG reconstruction, DDG extraction, and symbolic execution—we observe consistent failure patterns across various LLMs. These failures suggest fundamental limitations in LLMs' ability to reason over IRs, reflecting broader architectural or training shortcomings rather than artifacts of dataset size.

While we agree that future work should expand to larger and more diverse datasets, our findings provide strong empirical evidence of systemic challenges in IR-level reasoning, even in simplified and controlled scenarios.

## C. Prompts

We design our prompts to align with the requirements of each IR task. In particular, we adopt two main prompting strategies:

- **Task-Specific Expert Prompts.** For each task, we assign a specialized role to the model (e.g., "CFG analyzer for IRs") and clearly define the input, task objective, and expected output format. For example, in Task 1, the output must follow DOT format with explicit node and edge definitions.
- **Advanced Prompting Techniques.** We employ *zero-shot* and *few-shot prompting* in Tasks 1–3 to provide context or examples where needed. For Task 4, we further incorporate *Chain-of-Thought (CoT) prompting* to encourage step-by-step execution reasoning.

### C.1. Prompt for Structural Understanding: Inferring Control Flow from IRs (Task 1)

| Prompt |
| --- |
| You are a control flow graph analyzer for Intermediate Representations (IRs). I will provide you with LLVM Intermediate representation (IRs), a low-level, platform-independent representation of a program. 
 Here is the IR code input: **[IR]** 
 Your task is to generate the control flow graph from the IR. The output format should be a DOT file, including nodes and edges. You do not need to list the content of each basic block; show each node's title. 
 Here is the IR code example to follow: **[IR Example]** 
 The output of the control flow graph should exactly match the following format: **[CFG Example]** |

### C.2. Prompt for Syntactic Comprehension: Decompiling IRs to High-Level Code (Task 2)

| Prompt |
| --- |
| You are an expert in high-performance computation. I will provide you with LLVM IRs (Intermediate Representations), which is a low-level, platform-independent representation of a program. 
 Here is the IR code input: **[IR]** 
 Your task is to decompile this IR code into a pure C or C++ source code format that can be run directly. 
 **Do not add any extra comments, explanations, or characters, and do not use any markdown formatting like "' or "'cpp.** |

### C.3. Prompt for Semantic Comprehension: Generating Natural Language Descriptions from IRs (Task 3)

| Prompt |
| --- |
| I will give you an IR code. Here is the IR code input: **[IR]** 
 I would like you to summarize the code according to the following specifications: 1. **Output Format**: - There are [n] functions in the code: - function[I (I in n)] takes [m] inputs: input1, input2, ..., inputm. - Function[I (I in n)] is [doing semantical function] on input1, input2, ..., inputm, and outputs [output]. 
 2. **Type and Variable Name Mapping Rules**: - '% "class.std::vector"*' should be summarized as 'vector¡float¿ numbers'. - Other IR-specific types should be mapped to their equivalent C++ types, following this pattern where possible. 
 3. **Strict Adherence to Formatting**: - The summary should strictly match the format provided below. - No additional comments, explanations, or deviations from the format should be added. - Do not use any markdown formatting such as "' or "'cpp. 
 Here is the IR code example to follow: **[IR Example]** 
 The output should exactly match the following format: **[Output Example]** 
 **Important Instructions**: - The summary must not include any additional comments, explanations, or formatting. - Ensure that variable names and types are directly transcribed as described in the example. - No markdown formatting (e.g., no "' or "'cpp) should be used. **Do not deviate from the specified format under any circumstances.** |

### C.4. Prompt for Execution Reasoning: Inferring Program Behavior (Task 4)

> **Prompt**
>
> I need your help to analyze whether a given assertion passes or fails based on the provided LLVM IR code for a function.
> LLVM IR Code: **[IR]**
> Assertion Statement: **[Assertion]**
> Please write down your thinking process, and list the pass/fail result of each assertion at the end. NOTE: Make sure the format the pass/fail result of each assertion at the end follow the example: **[Output Example]**

## D. HumanEval Setups

Building on prior work (Zheng et al., 2023; Tan et al., 2024), we utilize widely recognized benchmarks, specifically HumanEval and its extended test case version, to evaluate the ability of five state-of-the-art LLMs to comprehend semantics and compare them against various golden baselines. HumanEval, introduced by OpenAI, is a benchmark designed to assess the multilingual capabilities of code-generative models. It comprises 164 carefully handwritten programming challenges, each featuring a function signature, a natural language (NL) description, a function body, and a set of unit tests, with an average of 7.7 assertion statements per challenge (Chen et al., 2021). In our experiments, these 164 C++ programs serve as the source code for IR analysis.

The compilation experiments were conducted on a Dell Workstation equipped with 32 Intel(R) Xeon(R) CPUs E5-2620 v4 @ 2.10GHz, running on an x86-64 architecture with a 64-bit system. For these experiments, we used Clang adapted for LLVM 13 on Ubuntu 18.04. The C++ source code were compiled into IRs (.bc files) using the following command:

```
clang++ -O{Opt-level} -emit-llvm {benchmark}.cpp -S -o {benchmark}.bc
```

We use LLVM's built-in passes to generate golden CFGs with the *-dot-cfg* option to ensure accurate and consistent representations of the code structure. The golden code summarization is meticulously performed to ensure that the generated graphs and code representations faithfully reflect the original source code's semantics. This process is critical for precise evaluations in subsequent tasks, such as static analysis, code summary analysis, and dynamic analysis. By employing these methodologies, we ensure that the golden datasets are of the highest fidelity, providing a solid foundation for assessing the granular understanding and representation of IRs by LLMs.

## E. Repetition Results for Stability Analysis

To evaluate the stability of model performance, we repeated each experiment three times (Run 1–Run 3). The results showed minimal variance across runs, demonstrating the robustness of our evaluation. The following tables summarize the performance of DeepSeek R1 across repetitions.

### E.1. CFG Construction (Task 1)

Table 6 presents three runs of our CFG reconstruction task by DeepSeek R1 alongside GPT-4's performance. It demonstrates the consistency of our method across repeated trials and highlights performance relative to a state-of-the-art baseline.

*Table 6.* CFG construction performance across three DeepSeek R1 runs and GPT-4.

| Run | Completed | Node Accuracy | Full Accuracy | Partial Accuracy |
|-----|-----------|---------------|---------------|------------------|
| Run 1 | 69 | 55 | 53 | 2 |
| Run 2 | 77 | 64 | 57 | 6 |
| Run 3 | 73 | 62 | 60 | 2 |
| GPT-4 | 164 | 50 | 39 | 11 |

### E.2. IR Decompilation (Task 2)

Table 7 compares decompilation task results over three DeepSeek R1 runs, focusing on re-execution correctness. Llama 3.1 is included for reference. The data shows slight variation across runs, but consistent overall trends.

Table 7. IR decompilation performance across three DeepSeek R1 runs and Llama 3.1.

| Model | Completed | Re-execution Completed | Re-execution Success |
|---|---|---|---|
| Run 1 | 72 | 36 | 18 |
| Run 2 | 77 | 38 | 17 |
| Run 3 | 75 | 39 | 14 |
| Llama 3.1 | 77 | 23 | 14 |

### E.3. Code Summarization (Task 3)

Table 8 shows the summarization performance of three DeepSeek R1 runs across BLEU, ROUGE, and METEOR metrics. It includes both high-quality summary counts ($> 0.8$) and average scores per run. Llama 3.1 is listed for comparison.

Table 8. Code summarization results across three DeepSeek R1 runs and Llama 3.1.

| Model | Task Comp. | > 0.8 BLEU | ROUGE | METEOR | BLEU | ROUGE | METEOR |
|---|---|---|---|---|---|---|---|
| Run 1 | 49 | 1 | 5 | 8 | 0.413 | 0.637 | 0.699 |
| Run 2 | 49 | 1 | 5 | 10 | 0.420 | 0.639 | 0.692 |
| Run 3 | 49 | 1 | 4 | 10 | 0.433 | 0.640 | 0.705 |
| Llama 3.1 | 81 | 1 | 5 | 11 | 0.390 | 0.610 | 0.670 |

### E.4. Execution Reasoning (Task 4)

Table 9 presents execution reasoning results across three DeepSeek R1 runs, covering both IR-based and source-code-based variants. It includes Pass/Partial Pass counts and final pass rates. Llama 3.1 is added for reference.

Table 9. Execution reasoning results on IR and source code across three independent DeepSeek R1 runs, compared with Llama 3.1.

| Model | IR Comp. | SC Comp. | IR Pass | SC Pass | IR Partial | SC Partial | IR Rate | SC Rate |
|---|---|---|---|---|---|---|---|---|
| Run 1 | 164 | 164 | 31 | 133 | 133 | 31 | 0.189 | 0.811 |
| Run 2 | 164 | 164 | 30 | 146 | 134 | 18 | 0.183 | 0.817 |
| Run 3 | 164 | 164 | 32 | 139 | 132 | 25 | 0.195 | 0.804 |
| Llama 3.1 | 164 | 164 | 31 | 119 | 114 | 35 | 0.189 | 0.720 |

## F. Cases of Findings

### F.1. Example of CFG construction

Figure 4 illustrates an end-to-end example of DeepSeek's Chain-of-Thought (CoT) reasoning for control flow graph (CFG) generation from LLVM IR (with the IR code shown in Figure **??**). The model receives LLVM IR code as input, produces an explicit multi-step reasoning trace to identify the control flow structure, and finally outputs the corresponding DOT file representation of the CFG.

### F.2. Example of decompilation

Figure 5 shows a re-execution mismatch from GPT-4. The original code searches for a value in the `key` object and increments `out` when not found. In contrast, the decompiled version checks only for matching characters in two strings, incrementing `count` instead. This discrepancy likely arises because GPT-4 loses context during loop reconstruction, leading to partial logic and a different outcome during re-execution.

---

**DeepSeek Chain-of-Thought Reasoning and Output**

**Step-by-step reasoning:**

Okay, so I'm trying to create a control flow graph (CFG) for the given IR code. The code is in LLVM IR format, and the function is named _Z15truncate_numberf. Let me break this down step by step.

First, I need to understand what the function does. It takes a float as input and returns a float. The function seems to involve some floating-point to integer conversion and back. Let's look at the instructions:

1. %2 = fptosi float %0 to i32: This converts the input float %0 to a 32-bit integer and stores it in %2.
2. %3 = sitofp i32 %2 to float: Then, this integer %2 is converted back to a float and stored in %3.
3. %4 = fsub float %0, %3: This subtracts %3 from %0, storing the result in %4.
4. ret float %4: Finally, the result %4 is returned.

Now, to create the CFG, I need to identify the basic blocks. A basic block is a sequence of instructions without any control flow changes. In this function, there are no branches or loops, so it's a straight-line sequence of instructions. That means there's only one basic block.

The start node is the entry point of the function, which is the beginning of the function where the arguments are prepared. Then, each instruction follows in sequence. The end node is the return statement where the function exits.

So, the CFG will have three nodes: the start node, the basic block containing all the instructions, and the end node. The edges will show the flow from start to the basic block and then to the end.

I need to represent this in a DOT file format. Each node will be a label, and the edges will be directed arrows showing the flow from one node to another.

Putting it all together, the DOT file will have nodes labeled as start, the basic block, and end, with edges connecting them in order.

**Generated DOT output:**

```
digraph CFG {
    node [shape=box, style=filled];

    start [label="Start"];
    basic_block_1 [label="Basic Block"];
    end [label="End"];

    start -> basic_block_1;
    basic_block_1 -> end;
}
```

*This graph shows the sequential execution from the start, through the single basic block, to the end.*

*Figure 4.* An example of DeepSeek's step-by-step reasoning and DOT output for CFG generation from LLVM IR.

Original source code

```
1     for (int i=0; i<num.length();i++)
2         if(find(key.begin(), key.end
              (), num(i) != key.end))
                  out+=1;
```

Code from decompilation

```
1     for (size_t i=0; i<length;++i){
2         if (i<tmp.length() && str[i] ==
              tmp[i])
3             count++; }
```

*Figure 5.* An example of re-execution mismatch from GPT-4.

## F.3. Example of summarization

Figure 6 illustrates the LLVM Intermediate Representation (IR) of the `truncate_number` function, which extracts the decimal part of a floating-point number.

Table 10 presents the summarization of the Example IR generated by various LLMs, showing how each model interprets the function's behavior.

```
1  ; ModuleID = '../../Human_xBenchmarks_164/CPP_2/CPP_2.cpp'
2  source_filename = "../../Human_xBenchmarks_164/CPP_2/CPP_2.cpp"
3  define dso_local float @_Z15truncate_numberf(float %0) {
4    %2 = fptosi float %0 to i32
5    %3 = sitofp i32 %2 to float
6    %4 = fsub float %0, %3
7    ret float %4
8  }
```

LLVM IR Source Code

*Figure 6.* An example of LLVM IR.

*Table 10.* Summarization of example IR by different LLMs.

| LLMs | Summarization of Example IR |
|------|------------------------------|
| **Ground Truth** | There is 1 function in the code. Function[1] takes 1 input: a float number. It truncates the float number to its integer part and outputs decimal part of the floating-point number. |
| **GPT-4** | There is 1 function in the code. Function[1] takes 1 input: a float number. It truncates the float number to its integer part and outputs the fractional part as a float. |
| **GPT-3** | There is 1 function in the code. Function[1] takes 1 input: a float number. It truncates the decimal part of the input number and outputs the truncated value. |
| **Gemma 2** | There is 1 function in the code. Function[1] takes 1 input: a float number. It truncates the fractional part of the input number and outputs a float. |
| **Llama 3.1** | There is 1 function in the code. Function[1] takes 1 input: a float number. It truncates the decimal part and outputs the decimal part of the input number. |
| **Code Llama** | There is 1 function in the code. Function[1] takes 2 inputs: a vector of floats and a float threshold. It checks if any two elements in the vector are closer to each other than the threshold and outputs a boolean value indicating if such elements exist. |

## F.4. Example of execution reasoning

Fig. 7 presents the source code for execution reasoning. The IR code shown is the same as the one used for summarization in the previous section.

*Table 11.* Execution reasoning result of the IR example from GPT-4.

| assert (truncate_number(3.5) == 0.5); → **Pass** | |
|---|---|
| **Step 1: Understanding the Function** | The function `truncate_number` takes a floating-point number as input. It uses `int()` to truncate the decimal part, returning the largest integer smaller than or equal to the input. Then, it subtracts this integer from the original number to get the decimal part. |
| **Step 2: Analyzing the Input** | The assertion checks if `truncate_number(3.5)` equals `0.5`. Let's evaluate it step-by-step:

• `int(3.5)` returns `3`

• `3.5 - 3` equals `0.5` |
| **Step 3: Comparing with the Assertion** | The assertion compares the function output with the expected value:

• Since `truncate_number(3.5)` returns `0.5`, and the assertion expects `0.5`, the result is `true`. |
| **Step 4: Conclusion**

✓ **Assertion Result: Pass** | The assertion passes because the expected value (`0.5`) matches the actual output of the function (`0.5`). |

*Table 12.* An example of execution reasoning result of the C++ example from GPT-4.

| assert (truncate_number(3.5) == 0.5); → **Pass** | |
|---|---|
| **Analysis of the Provided C/C++ Code** | The function `truncate_number` takes a floating-point number as input and returns the decimal part. It uses `int()` to truncate the number, effectively removing the fractional part. The decimal part is then calculated as the original number minus the integer part. |
| **Step-by-Step Breakdown** | • The input `number` is passed to the function.

• `int(number)` truncates the decimal part.

• The truncated integer is subtracted from the original `number`.

• The result (decimal part) is returned. |
| **Analysis of the Assertion Statement** | The assertion checks whether `truncate_number(3.5)` equals `0.5`. Evaluation steps:

• `int(3.5)` gives `3`.

• `3.5 - 3` equals `0.5`.

• So, the function should return `0.5`. |
| **Final Result**
✓ **Assertion Result: Pass** | The actual output matches the expected result. |

```
   Code from execution reasoning

1      #include <stdio.h>
2      #include <math.h>
3      float truncate_number(float number)
           {
4          return number - (int)number;
5      }
```

*Figure 7.* An example of source code C++.

F.4.1. MISUNDERSTANDING ALGORITHM LOGIC (45 CASES)

**Example (CPP_69):**

```
assert (search({5, 5, 5, 5, 1}) == 1);
```

**Expected behavior:** The function should find the greatest integer with a frequency greater than or equal to its value.
**Example (CPP_121) as shown in Table 13:**

```
assert (solutions({5, 8, 7, 1}) == 12);
```

**Expected behavior:** Sum all odd elements at even positions.
**LLM failure:** Instead of following the logic step-by-step, the model guesses the sum based on pattern recognition from prior examples rather than executing the correct loop iteration. **LLM failure:** The model fails to track frequency accumulation and instead returns an incorrect value, misunderstanding how the search function should operate.

F.4.2. PATTERN-BASED GUESSING (35 CASES)

**Example (CPP_121) shown in Table 13:**

```
assert (solutions({5, 8, 7, 1}) == 12);
```

**Expected behavior:** Sum all odd elements at even positions.
**LLM failure:** Instead of following the logic step-by-step, the model guesses the sum based on pattern recognition from prior examples rather than executing the correct loop iteration.

F.4.3. CONTROL FLOW ISSUES (28 CASES)

**Example (CPP_92):**

```
assert (any_int(2.5, 2, 3) == false);
```

**Expected behavior:** The function should correctly handle nested conditionals to determine if any sum of two numbers equals the third.
**LLM failure:** The model does not properly evaluate the if-else branching and incorrectly processes non-integer values.

F.4.4. OVERRELIANCE ON FUNCTION NAMES (22 CASES)

**Example (CPP_105):**

```
assert (issame(by_length({2, 1, 1, 4, 5, 8, 2, 3}) ,
{"Eight", "Five", "Four", "Three", "Two", "Two", "One", "One"}));
```

**Expected behavior:** Sort numbers, reverse the list, and replace them with corresponding word representations.
**LLM failure:** Instead of applying the transformation rules, the model simply assumes the function operates based on its name and produces an incorrect output.

*Table 13.* An example of guessing result based on patterns from GPT-4.

| assert (solutions(5, 8, 7, 1) == 12); → **Fail** | |
|---|---|
| **Step 1: function signature** | the function 'solutions' takes a pointer to a '$std :: vector < int >$' as its argument. the vector is expected to contain integers. |
| **Step 2: loading vector elements** | - the function uses 'getelementptr' and 'load' instructions to access the elements of the vector.
- the vector is accessed in a way that suggests it is iterating over its elements. |
| **Step 3: vector operations** | - the function performs several operations on the elements of the vector, including loading elements, performing shifts, and using the 'srem' (signed remainder) operation.
- the function also uses a vector reduction operation ('@llvm.vector.reduce.add.v4i32') which indicates that it sums up the elements of the vector in groups of four. |
| **Step 4: understanding the logic** | - the function appears to be summing the elements of the vector, but it also applies some conditions based on the results of the 'srem' operations.
- the 'srem' operation checks if the elements are odd (i.e., remainder when divided by 2 is 1) and conditionally includes them in the sum. |
| **Step 5: input values** | - the input vector '5, 8, 7, 1' contains the integers 5, 8, 7, and 1.
- the odd numbers in this vector are 5, 7, and 1. the even number is 8. |
| **Step 6: calculating the expected result** | - the sum of all elements is '5 + 8 + 7 + 1 = 21'.
- however, since the function seems to apply a filter based on the 'srem' operation, we need to check which elements are included in the final sum.
- the 'srem' operation checks if the remainder is 1, which means only the odd numbers (5, 7, and 1) would be included in the sum.
- therefore, the sum of the odd numbers is '5 + 7 + 1 = 13'. |
| **Step 7: final comparison** | - the assertion checks if the result of 'solutions(5, 8, 7, 1)' equals 12.
- based on our analysis, the function would return 13, not 12. |

F.4.5. STRING PROCESSING ERRORS (18 CASES)

**Example (CPP_112):**

```
assert (issame(reverse_delete("abcde","ae") , {"bcd","False"}));
```

**Expected behavior:** Remove characters and check for palindromicity.
**LLM failure:** The model fails to correctly apply character deletions and check the reverse consistency, leading to incorrect assertions.

**F.5. Conclusion**

These findings highlight common failure patterns in LLM-based code interpretation, including logical misinterpretation, heuristic-based assumptions, and issues with numerical computations. Further refinement in model training and debugging processes is recommended.

