# OpenReview forum: "Can Large Language Models Understand Intermediate Representations in Compilers?"
_ICML.cc/2025/Conference — ICML 2025 poster_

### Official Review · Reviewer_ua9h · 2025-03-09

**Overall Recommendation:** 3

**Summary:**

This paper presents an empirical study of the capability of LLMs to understand intermediate representations (IRs) of code. The LLMs are evaluated on 4 types of tasks of IR understanding: control-flow graph (CFG) reconstruction, IR decompilation, code summarization and execution reasoning. The results indicate that while LLMs are capable of parsing IR syntax and recognizing high-level structures in code tasks, they struggle with control flow reasoning, execution semantics, and loop handling.

**Claims And Evidence:**

The paper provides experiments to support the claims, by using HumanEval dataset and five LLMs.

**Essential References Not Discussed:**

No.

**Experimental Designs Or Analyses:**

The designs and analysis of experiments make sense for all 4 types of tasks, demonstrating the LLMs’ capability of understanding IRs.

The evaluation of LLMs is usually conducted in multiple turns, and the results are often presented with statistical metrics such as pass@k, while whether the results in this paper are obtained in multiple turns is not specified.

**Methods And Evaluation Criteria:**

The method is only evaluated on HumanEval dataset. This dataset is at the code function level with the average code lines of less than 10.

The paper investigates GPT-4, GPT-3, Gemma 2, LLaMA 3.1, and Code Llama, in understanding IRs. Since the capabilities of LLMs are evolving rapidly, SOTA LLMs should be used for evaluation.

**Other Comments Or Suggestions:**

No.

**Other Strengths And Weaknesses:**

Strengths:
1. This paper presents a empirical study to investigate the capabilities of LLMs, including GPT-4, GPT-3, Gemma 2, LLaMA 3.1, and Code Llama, in understanding IRs.
2. The IR comprehension of LLMs is analyzed across four tasks: Control Flow Graph (CFG) reconstruction, decompilation, code summarization, and execution reasoning.

Weaknesses:
1. The paper only uses HumanEval dataset, which is at code functional levels with fewer average lines.
2. The paper evaluates the capabilities of LLMs, including GPT-4, GPT-3, Gemma 2, LLaMA 3.1, and Code Llama. More SOTA LLMs should be included, such as GPT-4o, DeepSeek-V3 or R1, Qwen.
3. There are non-neural developed tools for these tasks, many of them integrated in commonly-used IDEs or software testing applications. If the performances of these tools can be listed together, they could show the advantages and disadvantages of LLM-based methods.
4. The evaluation of LLMs is usually conducted in multiple turns, and the results are often presented with statistical metrics such as pass@k, while whether the results in this paper are obtained in multiple turns is not specified.

**Questions For Authors:**

Please refer to the weaknesses.

**Relation To Broader Scientific Literature:**

This paper lists multiple relevant papers and analysis their differences. For example, Meta’s LLM Compiler offers pre-trained models for code optimization tasks. While prior work has explored IR representation learning for code optimization and analysis, no studies have systematically examined how LLMs comprehend IR syntax, CFG structures, execution behavior, and semantic relationships. This paper addresses this gap by providing the first empirical evaluation of LLMs’ IR comprehension across these dimensions.

**Theoretical Claims:**

This paper is an empirical study. There is no proof of theoretical claims.

---

> ### Author Rebuttal · Authors · 2025-04-01
>
> We sincerely thank the reviewer for the thoughtful and constructive feedback. Below, we address the key concerns:
>
> **W1:** The paper only uses the HumanEval, which contains code functions with an average of fewer than 10 lines.
>
> **A1:** Though HumanEval consists of relatively short functions, our results reveal fundamental limitations in LLMs’ IR comprehension that extend beyond dataset scale from three aspects: **(1) Controlled Complexity:** Even small programs require precise reasoning about IR constructs; **(2) Task Difficulty Beyond Code Size:** Despite their brevity, small programs exhibit complex low-level semantics.; **(3) Systemic Failures:** Consistent errors across tasks—such as CFG Construction and Execution Reasoning—indicate systemic deficiencies in IR understanding.
>
> ---
>
> **W2:** The paper evaluates GPT4, Gemma, and so on. More SOTA LLMs, such as GPT-4o, DeepSeek-V3 (or R1), should be included.\
> &\
> **W4:** It is not specified whether the results are obtained in multiple turns.
>
> **A2:** We agree that including SOTA models is essential. We supplemented our experiments with DeepSeek R1.
>
> **A4:** We fully agree that multi-turn evaluations using metrics like pass@k enhance robustness. We ran DeepSeek R1 on all four tasks three times (R1–R3) to ensure stability.
>
> T1: CFG Construction
>
> |       | Comp. | Node Acc. | Full Accu. | Partial Accu. |
> |:-:|:-:|:-:|:-:|:-:|
> | R1 |     69     |     **55**    |     53     |       2       |
> | R2 |     77     |     **64**    |     57     |       6       |
> | R3 |     73     |     **62**    |     60     |       2       |
> | GPT-4 |     164    |     50    |     39     |       11      |
>
> T2: IR Decompilation
>
> |       | Comp. | Re-exe. Comp. | Re-exe. Success |
> |:-:|:-:|:-:|:-:|
> | R1 |       72       |           36           |          18         |
> | R2 |       77       |           38           |          17         |
> | R3 |       75       |           39           |          14         |
> |LLaMA 3.1|    77      |          23            |          14         |
>
> T3: Code Summarization
>
> |           | Task Comp. | BLEU > 0.8 | METEOR > 0.8 | ROUGE > 0.8 |  Avg. BLEU |  Avg. ROUGE |  Avg. METEOR |
> |:-:|:-:|:-:|:-:|:-:|:-:|:-:|:-:|
> |   R1   |     49     |   1   |   5   |    8   | 0.413 | 0.637 |  0.699 |
> |   R2   |     49     |   1   |   5   |   10   | 0.420 | 0.639 |  0.692 |
> |   R3   |     49     |   1   |   4   |   10   | 0.433 | 0.640 |  0.705 |
> | LLaMA 3.1 |     81     |   1   |   5   |   11   |  0.39 |  0.61 |  0.67  |
>
> T4: Execution Reasoning
>
> |       | IR Com. |SC Com. | IR Pass | SC Pass | IR Partial Pass |SC Partial Pass | IR Pass % | SC Pass % |
> |:-:|:-:|:-:|:----:|:-:|:-:|:-:|:-:|:-:|
> | R1 |       164      |       164      |  31  |  133 |      133     |      31      |   18.9   |   81.1   |
> | R2 |       164      |       164      |  30  |  146 |      134     |      18      |   18.3   |   81.7   |
> | R3 |       164      |       164      |  32  |  139 |      132     |      25      |   19.5   |   80.4   |
> |LLaMA 3.1 |       164      |       164      |  31  |  119 |      114     |      35      |   18.9   |    72.0   |
>
> DeepSeek R1 performs comparably to LLaMA 3.1 on T2–T4 but excels in T1 (CFG Construction) due to its integrated chain-of-thought mechanism that helps identify basic blocks and critical control flow instructions (e.g., "br" and "jmp").
>
> **Revised Finding 1:** LLMs generally struggle with detecting basic blocks and constructing control flow edges; however, chain-of-thought prompting modestly enhances the recognition of key control flow instructions.
>
> **A4:** Notably, the three runs yielded consistent results, demonstrating the reliability of DeepSeek R1’s performance. We will include these multi-turn results in the revised manuscript.
>
> **In the final version, we will include comprehensive analyses, additional experiments, and concrete examples to support these findings.** In future work, we will systematically explore additional prompting techniques to further validate and extend these findings.
>
> ---
>
> **W3:** There exist non-neural developed tools for IR tasks in common software testing applications that could help highlight the dis/advantages of LLM-based methods.
>
> **A3:** We appreciate the reviewer’s point that established non-neural methods serve as valuable baselines. Tools such as Ghidra, various IDE tools, and symbolic execution engines like KLEE excel at CFG construction, decompilation, and execution reasoning through extensive domain-specific training. In contrast, our study uses these tasks solely as diagnostic benchmarks to evaluate the raw, untuned IR comprehension of LLMs. Our goal is not to outperform specialized tools but to reveal intrinsic LLM limitations—such as in control flow inference, granular semantic understanding, and loop handling—that can guide future optimizations. In the final manuscript, we will include a detailed discussion comparing our results with these non-neural baselines.
>
> ---
>
> W4 & A4 are included in A2.

---

> > ### Comment · Reviewer_ua9h · 2025-04-08
> >
> > Thanks for the clarification. I have updated the score.

---

> > > ### Author Response · Authors · 2025-04-08
> > >
> > > Thank you very much for your updated evaluation and for taking the time to review our rebuttal. We greatly appreciate your feedback and value your insights into our work. Your constructive comments are crucial for us, and we welcome any further concerns or suggestions you may have as we continue to refine our research.
> > >
> > > Best Regards,
> > >
> > > The Authors

---

### Official Review · Reviewer_2sD2 · 2025-03-12

**Overall Recommendation:** 3

**Summary:**

The paper experiments with applying LLMs to the control flow graph of programming code, identifying key challenges of control flow, semantic understanding, and loop handling. These challenges, as analyzed through 4 tasks, seem to permeate over a variety of language models including Coda Llama, Gemma 2, and GPT-4. The paper claims to be the first to analyze how LLMs perform on intermediate representations of the compiler.

**Claims And Evidence:**

The main claim is negative, that LLMs struggle with CFG-based tasks. This negative claim receives support from the results, with some limitations (as described below) -- as common with negative claims -- where a comprehensive study of all possibilities is difficult to bring forward. It is difficult to put these results that the LLMs currently achieve into context without seeing some non-LLM baselines for these specific tasks.

**Essential References Not Discussed:**

None that I'm aware of, yet it is not well argued why methods from the related work were not considered as baselines.

**Experimental Designs Or Analyses:**

The experimental design seems to be appropriate for the studied research question of how LLMs would perform on CFG-related tasks.

**Methods And Evaluation Criteria:**

The dataset is a processed version of a subset of MMLU, which is an established benchmark for evaluating LLMs. The tasks around the CFGs are newly designed and seem to make sense. There are technically no new proposed methods, but a new application of LLMs to these newly designed tasks.

**Other Comments Or Suggestions:**

- The paper could be strengthened by efforts to improve the basic LLM performance with well-established basic prompting techniques (e.g., chain-of-thought, few-shot prompting)
- The paper could be improved by adding task-specific baselines.

**Other Strengths And Weaknesses:**

## Strengths

- The paper provides an in-depth analysis of applying LLMs to control flow graphs
- The importance of analyzing control flow graphs is nicely motivated.
- Specific findings are emphasized and justified by the results.
- Applying LLM to the intermediate representations of a compiler is, to the best of my knowledge, a novel application of LLMs

## Weaknesses

- The paper doesn't consider any non-LLM baselines, e.g. graph neural nets (as those mentioned in the related work) that would be arguably are more natural fit for the task. Even if LLMs currently don't compete with GNNs and other graph representation learning methods, it would be interesting to see how big the margin is.
- Evaluation is limited to zero-shot experiments. There are no few-shot experiments and no basic CoT prompting was tried. Both these are fairly established as standard techniques and could very well influence the conclusions.

**Questions For Authors:**

Are there any non-LLM approaches that would be applicable to the introduced tasks, e.g., fine-tuning or training a classifier on top of IR2vec / Bert-style models and/or applying graph representation learning methods? Or is there any particular reason that those are not included/applicable?

**Relation To Broader Scientific Literature:**

The key contributions match with the general trend of applying LLMs to a variety of problems. The paper covers a particular topic within applying LLMs to code.

**Theoretical Claims:**

Not applicable.

---

> ### Author Rebuttal · Authors · 2025-04-01
>
> We thank the reviewer for the thoughtful feedback and recognition of our novel application of LLMs to compiler IRs. We also appreciate the acknowledgement of  our focus on task-specific analysis. We address the key concerns below.
>
> **Q1 & W1:** Are there any non-LLM approaches that would be applicable to the introduced tasks, e.g., fine-tuning or training a classifier on top of IR2vec / Bert-style models and/or applying graph representation learning methods? Or is there any particular reason that those are not included/applicable\
> &\
> **W3:** The paper does not include non-LLM baselines.
>
> **A1:** We appreciate the reviewer’s point that established non-LLM methods can serve as valuable baselines. For instance, pre-trained models such as FAIR [1] have shown impressive results in semantic summarization. However, while non-LLM approaches (e.g., GNNs, fine-tuned BERT-style models) excel on specific tasks, they require extensive dataset-specific training and are typically applied to higher-level applications.
>
> In contrast, our study focuses on the raw, untuned IR comprehension of general-purpose LLMs. By using tasks such as CFG reconstruction, IR decompilation, code summarization, and execution reasoning, we establish diagnostic benchmarks that evaluate the zero/few-shot and chain-of-thought (CoT) prompting capabilities of current LLMs without any task-specific fine-tuning. This approach not only reveals the inherent limitations of existing LLMs in understanding intermediate representations but also provides actionable insights for future improvements.
>
> Moreover, as detailed in **A2**, we will incorporate additional CoT prompting in the revised manuscript. In future work, we plan to explore non-LLM baselines and a broader range of prompting techniques to further enhance our analysis.
>
>
>
>
> ---
>
> **W2 & W3:** Evaluation is limited to zero-shot experiments; few-shot and basic CoT prompting have not been explored.
>
> **A2:** Our evaluation is not strictly zero-shot. We use zero-shot prompting for CFG Construction and IR Decompilation, few-shot examples for Code Summarization, and chain-of-thought prompting for Execution Reasoning. Although these techniques improve performance, fundamental challenges in IR understanding—particularly in control flow and execution semantics—persist, with model rankings remaining unchanged.
> To assess the benefits of chain-of-thought prompting, we conducted supplementary experiments using DeepSeek R1. We selected DeepSeek R1 because (1) it is among the state-of-the-art LLMs, and (2) it features an inherent chain-of-thought mechanism that decomposes complex IR tasks into intermediate reasoning steps, showing potential advantages for our IR tasks.
>
> We ran DeepSeek R1 on all four tasks three times to ensure stability. Our preliminary results (one as an example) are as follows:
>
> T1: CFG Construction
>
> |       | Comp. | Node Acc. | Full Accu. | Partial Accu. |
> |:-:|:-:|:-:|:-:|:-:|
> | R2 |     77     |     **64**   |     57     |       6       |
> | GPT-4 |     164    |     50    |     39     |       11      |
>
> T2: IR Decompilation
>
> |       | Comp. | Re-exe. Comp. | Re-exe. Success |
> |:-:|:-:|:-:|:-:|
> | R1 |       72       |           36           |          18         |
> |LLaMA 3.1|    77      |          23            |          14         |
>
> T3: Code Summarization
>
> |           | Task Comp. | BLEU > 0.8 | METEOR > 0.8 | ROUGE > 0.8 |  Avg. BLEU |  Avg. ROUGE |  Avg. METEOR |
> |:-:|:-:|:-:|:-:|:-:|:-:|:-:|:-:|
> |   R1   |     49     |   1   |   5   |    8   | 0.413 | 0.637 |  0.699 |
> | LLaMA 3.1 |     81     |   1   |   5   |   11   |  0.39 |  0.61 |  0.67  |
>
> T4: Execution Reasoning
>
> |       | IR Completed |SC Completed | IR Pass | SC Pass | IR Partial Pass |SC Partial Pass | IR Pass rate | SC Pass Rate |
> |:-:|:-:|:-:|:----:|:-:|:-:|:-:|:-:|:-:|
> | R1 |       164      |       164      |  31  |  133 |      133     |      31      |   0.189   |   0.811   |
> |LLaMA 3.1 |       164      |       164      |  31  |  119 |      114     |      35      |   0.189   |    0.72   |
>
> DeepSeek R1 performs comparably to LLaMA 3.1 on T2–T4 but excels in T1 (CFG Construction) due to its integrated chain-of-thought mechanism that helps identify basic blocks and critical control flow instructions (e.g., "br" and "jmp").
>
> **Revised Finding 1:** LLMs generally struggle with detecting basic blocks and constructing control flow edges; however, chain-of-thought prompting modestly enhances the recognition of key control flow instructions.
>
> **In the final version, we will include comprehensive analyses, additional experiments, and concrete examples to support these findings.** In future work, we will systematically explore additional prompting techniques to further validate and extend these findings.
>
> [1] Niu, Changan, et al. "Fair: flow type-aware pre-training of compiler intermediate representations." ICSE’24.

---

### Official Review · Reviewer_k99i · 2025-03-13

**Overall Recommendation:** 3

**Summary:**

The paper provides an empirical evaluation of current LLMs on IR understanding tasks, namely --

- CFG reconstruction
- decompilation
- code summarization, and
- execution reasoning

and find that models struggle with complex reasoning about IRs

**Claims And Evidence:**

- Pioneering empirical study to investigate the capabilities of LLMs -- first work to evaluate LLMs on IR related tasks
- Empirical findings
    - LLMs recognize syntax but struggle with control flow and semantics
    - Loop handling remains a fundamental challenge

These are demonstrated with comprehensive evaluations on the 4 tasks.

**Essential References Not Discussed:**

The related work for intermediate/assembly language is well provided, however, a rich body of work about code understanding is not discussed. For example [1] introduced code execution evaluations, which has been followed up by many other works attempting to use different static and dynamic approaches to evaluate code understanding (including cfg path analysis in [2])

[1] CRUXEval: Code Reasoning, Understanding, and Execution Evaluation
[2] LLMs: Understanding Code Syntax and Semantics for Code Analysis

**Experimental Designs Or Analyses:**

- The experimental design and evaluation are well presented. Each task is described clearly with appropriate metrics and prompts.

- Given IR understanding can be seen as an OOD task, it would be useful to further enhance the description about the prompting effort that was applied across different models.

- Reasoning models (such as O1 or R1) are not discussed. Given considerable performance gains witnessed on programming tasks, it deserves analysis in this paper.

**Methods And Evaluation Criteria:**

They propose new benchmarks (derived from compiling humaneval problems to IRs) on four diverse tasks to evaluate IR capabilities of models. The tasks are not necessarily novel and have been studied for programs in high-level language, but are new in the context of LLVM code.

The benchmark is collected from HumanEval problems, which might not provide insights toward understanding real-world programs. Particularly, given a challenge with IR / Assembly programs is terseness of such programs, it is unclear if those effects are accounted for here.

**Other Comments Or Suggestions:**

none

**Other Strengths And Weaknesses:**

Strength.
- The paper is well written and the findings are easy to understand. The recommendations for improving IR reasoning capabilities would also be useful for future work.

Weakness.
- Choice of HumanEval problems used for constructing tasks likely challenges generalizability of the findings.
- Current LLMs are likely not heavily optimized for IR programs. It is unclear how to calibrate the findings once we develop models explicitly optimized for IR programs. It is possible that the findings would considerably change for such models.

**Questions For Authors:**

- Can you provide more details on the level of prompting required for models to work with IR programs?

- Can you provide the statistics of the IR humaneval programs?

**Relation To Broader Scientific Literature:**

The paper contextualizes the research with relevant work in LLMs for intermediate/assembly language.

**Theoretical Claims:**

none

---

> ### Author Rebuttal · Authors · 2025-04-01
>
> We sincerely thank the reviewer for the thoughtful and constructive feedback. We appreciate your recognition of the novelty of applying LLMs to compiler IRs and the value of our in-depth evaluation. Below, we address the key concerns:
>
> **Q1:** Can you provide more details on the level of prompting?
>
> **A1:** In our paper, we have adopted two prompting strategies: **(1) Task-Specific Expert Prompts.**  For each IR task, prompts assign a role (e.g., “CFG analyzer for IRs”) and clearly define the input, task objective, and required output format (e.g., DOT files with nodes and edges); **(2) Advanced Prompting Techniques.** We also applied the *Zero/Few-Shot Prompting* and *Chain-of-Thought (CoT) Prompting* in Tasks 1–3 and Task 4, respectively.
>
> Note that the main text summarizes our approach, and complete prompt templates are provided in Appendix B (extended examples omitted for brevity).
>
> ---
>
> **Q2:** Can you provide the statistics of the IR humaneval programs?
>
> **A2:** Key statistics of 164 IR programs across optimization levels are as follows:
> | OPT Level | LoC | Tokens | Functions | Loops | Basic Blocks |
> |:-:|:-:|:-:|:-:|:-:|:-:|
> |   -O0     |    162,389    | 1,079,290 |   5,195   |  463  |   13,228     |
> |   -O1     |     66,598    |  454,954  |    393    |  577  |    7,708     |
> |   -O2     |     69,102    |  478,002  |    374    |  625  |    8,048     |
> |   -O3     |     75,519    |  526,395  |    367    |  827  |    8,917     |
>
> At -O0, minimal optimizations yield verbose IR with duplicate functions to preserve debugging (‘linkonce_odr’). -O1 and -O2 remove redundant elements, reducing size but slightly increasing loop counts, while -O3 further simplifies the structure at the cost of more loops. These trends explain why decompilation performs best at -O1/O2. We will include a detailed description and a summary table in the revised manuscript, along with data-driven explanations to substantiate our findings.
>
> ---
>
> **W1:** Choice of HumanEval problems used for constructing tasks likely challenges the generalizability.
>
> **AW1:** We acknowledge that the HumanEval dataset may limit generalizability. However, our results reveal “general limitations” in LLMs’ IR comprehension that extend beyond dataset scale or diversity from two aspects: **(1) Controlled Complexity:**  Although the programs are small, the extracted IRs involve common operations—including nested loops, conditional statements, and function calls—that pose significant challenges; **(2) Systemic Failures:** Consistent errors across tasks, such as CFG reconstruction and execution reasoning, indicate systemic deficiencies in IR understanding.
>
> These issues represent fundamental “general challenges” that LLMs have yet to overcome for most programs. Once these preliminary obstacles are addressed, we plan to expand our evaluations to larger, more diverse datasets to continuously enhance generalizability. We will also discuss future directions for incorporating more diverse IR samples in the revised manuscript.
>
>
> ---
>
> **W2:**  It is unclear how to calibrate the findings once we develop models explicitly optimized for IR programs
>
> **AW2:** We agree that fine-tuning on IR-specific datasets may improve performance. Our study is primarily exploratory, positioned to assess the untuned performance of current LLMs in understanding IRs without relying on benchmark-specific fine-tuning. We acknowledge that fine-tuning LLMs on specialized datasets could boost their performance on IR-related downstream tasks. However, by focusing on their untuned performance, our work establishes a clear baseline that exposes key challenges. Our findings pinpoint areas where current LLMs fall short—such as control flow comprehension (Finding 1), granular semantic understanding (Findings 4–5), and loop handling—while also offering guidance for targeted fine-tuning strategies. We are committed to pursuing fine-tuning as a central focus of our future research, leveraging our insights to significantly enhance models' abilities to tackle more complex IR-related tasks.
>
> ---
>
> **W3**: The related work for intermediate/assembly language is well provided; however, a rich body of work about code understanding is not discussed.
>
> **AW3:** We appreciate the suggestion to contextualize our work further within the broader code understanding literature. Although our manuscript comprehensively covers IR-level work, we agree that expanding the discussion of static/dynamic analysis methods (e.g., CRUXEval [2] and CFG path analysis in [3]) would enrich our related work section. In the revised manuscript, we will incorporate these references to better position our study within the broader landscape of code understanding research.
>
> [1] Chris C., et al. LLM Compiler: Foundation Language Models for Compiler Optimization. CC’25
>
> [2] CRUXEval: Code Reasoning, Understanding, and Execution Evaluation
>
> [3] LLMs: Understanding Code Syntax and Semantics for Code Analysis

---

### Official Review · Reviewer_k42A · 2025-03-13

**Overall Recommendation:** 4

**Summary:**

The authors explored the capabilities of large language models (LLMs) in understanding intermediate representations (IRs), primarily for applications such as code comprehension, optimization, and automated reasoning. Their findings indicate that while LLMs are proficient in understanding static IR features and basic control flow structures, they struggle with more complex representations, such as loop reasoning and execution simulation. Additionally, LLMs perform better at capturing semantic-level behavior rather than instruction-level details.

Their findings are highly significant for the broader research community in the field of LLMs, particularly in the domain of software analysis and handling IR-related tasks.

**Claims And Evidence:**

The claims and research findings are highly interesting and novel, supported by an extensive experimental setup and validated with convincing results. The findings are realistic and impressive.

**Essential References Not Discussed:**

NA

**Experimental Designs Or Analyses:**

I already mentioned and discussed in sections like "Methods And Evaluation Criteria*" and "Theoretical Claims*".

**Methods And Evaluation Criteria:**

Yes, the proposed methods, target study tasks/categories, and corresponding evaluation criteria are well-defined and clearly presented. The research is thorough and well-structured, providing an in-depth validation of the authors' claims and findings. The authors benchmarked the performance of selected but widely used LLMs, including GPT-3/4, Gemma 2, LLaMA 3.1, and Code Llama, on key IR-related tasks: (a) CFG construction, (b) decompilation ability, (c) code summarization, and (d) execution reasoning. Their findings effectively highlight the relative strengths and limitations of these LLMs for the aforementioned tasks.

**Other Comments Or Suggestions:**

several typos have been noticed,

**Other Strengths And Weaknesses:**

Strength:
1. The manuscript is well-written, thoroughly analyzed, and presents a detailed experimental setup to highlight the shortcomings of LLMs in reasoning and understanding complex IRs.
2. It provides a comprehensive benchmarking of widely used LLMs, including GPT-3/4, Gemma 2, LLaMA 3.1, and Code Llama, on key IR-related tasks: (a) CFG construction, (b) decompilation ability, (c) code summarization, and (d) execution reasoning.

Weakness:

The study does not consider fine-tuning on a benchmark-specific dataset, which limits its ability to reflect true benchmarking metrics. This omission is a drawback, as it does not fully showcase the potential capabilities of LLMs, particularly in the domain of software analysis.

**Questions For Authors:**

Please see Weakness and other comments

**Relation To Broader Scientific Literature:**

Their findings are highly significant for the broader research community in the field of LLMs, particularly in the domain of software analysis and handling complex IR-related tasks.

**Theoretical Claims:**

Yes, I have already elaborated on this in the "Methods and Evaluation Criteria" section. However, I also considered the research limitations highlighted by the authors. They acknowledged that their study is constrained by a limited benchmark dataset, as the HumanEval-derived IRs do not fully reflect data diversity. Additionally, the impact on model performance due to the lack of exploration into advanced prompting techniques and the omission of fine-tuning strategies—such as IR-specific dataset augmentation and fine-tuning—was not accounted for.

---

> ### Author Rebuttal · Authors · 2025-04-01
>
> Thank you very much for your detailed review and positive feedback on our work. We greatly appreciate your recognition of the novelty and significance of our study, as well as your thorough evaluation of our experimental setup. We would like to address your comments as follows:
>
>
> **W1:** The study does not consider fine-tuning on a benchmark-specific dataset, which limits its ability to reflect true benchmarking metrics.
>
> **AW1:** We appreciate the reviewer’s comment. Our study is primarily exploratory, positioned to assess the untuned performance of current LLMs (including zero/few-shot and chain-of-thought prompting) in understanding intermediate representations (IRs) without relying on benchmark-specific fine-tuning. This approach enables us to use tasks such as CFG reconstruction, IR decompilation, code summarization, and execution reasoning as benchmarks for evaluating LLMs' understanding of intermediate representations. Moreover, our findings reveal inherent limitations in how these models understand and process IRs.
> We acknowledge that fine-tuning LLMs on specialized datasets could boost their performance on IR-related downstream tasks. However, by focusing on their untuned performance, our work establishes a clear baseline that exposes key challenges. Our findings pinpoint areas where current LLMs fall short—such as control flow comprehension (Finding 1), granular semantic understanding (Findings 4–5), and loop handling—while also offering guidance for targeted fine-tuning strategies. We are committed to pursuing fine-tuning as a central focus of our future research, leveraging our insights to significantly enhance models' abilities to tackle more complex IR-related tasks.
>
> **Typos and Formatting Issues**
> We appreciate you highlighting some spelling and formatting errors. We have reviewed the manuscript carefully and will correct these issues in the final version.
>
> Again, thank you for your constructive comments.

---

### Decision · Program_Chairs · 2025-05-01

**Decision:**

Accept (poster)

**Comment:**

This paper presents an empirical study to investigate how well LLMs can understand Intermediate Representations (IRs) by evaluating them on tasks such as control flow graph reconstruction, decompilation, code summarization, and execution reasoning. The main results show that the evaluated LLMs can parse IRs and recognize high level structures, but struggle with more complex instruction level reasoning especially for branch and loop handling, as well as control flow reasoning and execution semantics. All reviewers found the study to be interesting and well executed, and important contribution to the research community. There were some concerns around limited number of SOTA LLMs (including reasoning ones) being evaluated as well as only using HumanEval problems potentially limiting the applicability to more real-world coding usecases. The response with additional experiments with R1 helped with some of these concerns. It would be great to include the discussions and experiments from rebuttal response to the final version of the paper.